# Brewer's Spent Grain with Yeast Amendment Shows Potential for Anaerobic Soil Disinfestation of Weeds and *Pythium irregulare*

Danyang Liu [1], Jayesh Samtani [1,*], Charles Johnson [2], Xuemei Zhang [2], David M. Butler [3] and Jeffrey Derr [1]

[1] Hampton Roads Agricultural Research and Extension Center, School of Plant and Environmental Sciences, Virginia Tech, Virginia Beach, VA 23455, USA; dadanyan@vt.edu (D.L.); jderr@vt.edu (J.D.)

[2] Southern Piedmont Agricultural Research and Extension Center, School of Plant and Environmental Sciences, Virginia Tech, Blackstone, VA 23824, USA; spcdis@vt.edu (C.J.); missizxm@vt.edu (X.Z.)

[3] Department of Plant Sciences, University of Tennessee, Knoxville, TN 37996, USA; dbutler@utk.edu

\* Correspondence: jsamtani@vt.edu; Tel.: +1-757-363-3901

**Abstract:** Anaerobic soil disinfestation (ASD) is a promising alternative to chemical fumigation for controlling soilborne plant pathogens and weeds. This study investigated the impact of brewer's spent grain (BSG), a locally available carbon source, on various weed species and the oomycete pathogen *Pythium irregulare* in ASD. Two greenhouse studies were conducted using BSG and yeast at full and reduced rates in a completely randomized design with four replicates and two runs per study. In both studies, ASD treatments significantly decreased the seed viability of all weed species and the *Pythium irregulare* inoculum, while promoting higher cumulative anaerobicity compared to the non-treated control. The addition of yeast had a notable effect when combined with BSG but not with rice bran. When used in reduced carbon rates, yeast supplementation enhanced the efficacy of BSG, providing comparable control to the full rate for most weed species, including redroot pigweed, white clover, and yellow nutsedge. Interestingly, no ASD treatment affected the soil temperature. Furthermore, BSG treatments caused higher concentrations of volatile fatty acids compared to ASD with rice bran and the non-treated control. This finding suggests that the inclusion of yeast in ASD shows potential for reducing the carbon input required for effective soil disinfestation.

**Keywords:** organic soil amendments; biological soil disinfestation; biological control; fumigant alternatives; fatty acids





## 1. Introduction

For many specialty crop producers that cultivate perennial or annual crops without crop or land rotation, the density of specific weeds, plant pathogens, and plant-parasitic nematode species can increase over time [1]. Pre-plant soil disinfestation strategies are useful and recommended for effective pest management on these farms. Growers have used pre-plant chemical fumigation of soil widely for decades, achieving efficient pest control and relatively high economic return compared to non-treated soils [2]. The chemical fumigants 1,3-dichloropropene (1,3-D) and chloropicrin have been used as alternatives to methyl bromide (MeBr) for pest control in several other horticultural crops [3]. However, chemical fumigation may potentially harm the environment and human health [4]. The use of improved barrier films on raised beds and expanding buffer zones to address environmental and human health concerns could further increase the cost of chemical fumigation [5]. Therefore, there is a need to develop and evaluate weed and crop-disease management alternatives to chemical fumigation.

Anaerobic soil disinfestation (ASD) involves using decomposable organic amendments (i.e., carbon sources to stimulate microbial respiration), irrigating to field capacity,

and covering with a polyethylene tarp to limit gas exchange and generate anaerobic conditions. Under anaerobic conditions, the facultative and obligate anaerobic microorganisms decompose the carbon sources, generating phytotoxic compounds which suppress multiple soilborne pathogens [6–8]. Several studies have reported the effectiveness of ASD for weed control; results have been variable among weed species, environments, soil temperatures, types and forms of carbon source, and field conditions [9,10]. For example, Florida studies comparing ASD using molasses and poultry litter as carbon sources to chemical fumigation for nutsedge control (*Cyperus* spp.) found ASD to be 85% more effective in reducing weed density compared to the nontreated control [11]. Herbicides applied with ASD have also been tested to address variability in weed control. Studies in Florida showed that halosulfuron-methyl (70 g/ha) improved the effectiveness of ASD for nutsedge control compared to ASD without this herbicide [12]. When molasses was used as a C source, ASD with fomesafen was ineffective in providing weed control relative to soil fumigation [13]. ASD with or without bioherbicides (Opportune™, 2.7 m/m$^2$, Marrone Bio Innovations Inc, Davis, CA, USA.) was not effective in the control of dicot weeds [14,15]. Although the pairing of ASD with herbicides may not consistently improve weed control, it indicated that it is possible to improve ASD efficacy in weed control by mixing with other amendments.

The types and rates of carbon sources are important components of ASD. The carbon sources should contain sufficient labile carbon and have a moderate carbon:nitrogen ratio to support soil microbial growth [16]. Currently, various labile carbon sources have been studied, such as molasses, rice bran (RB), wheat bran, ethanol, grass and other agricultural byproducts [1,16–20]. The liquid forms of carbon reported higher weed suppression efficacy compared to solid forms [9]. Among the liquid carbon forms, ethanol has shown a promising effect as a C source in ASD for controlling soilborne pests [16,19]. Ethanol is easier to apply and is more efficient in penetrating the soil compared to solid carbon sources [21]. However, the high cost of ethanol makes it unrealistic for large-scale usage in the USA, and ethanol is regulated when applied for agricultural use [22]. Instead, low-cost bioethanol produced from agricultural waste, or even byproducts from bioethanol fermentation, could potentially be used in ASD. In some studies [23,24], bioethanol could be made using local forage crops under field conditions. Moreover, Horita and Kitamoto [25] used the products and residue from bioethanol fermentation as carbon sources for ASD. Horita and Kitamoto concluded that the residue from bioethanol fermentation potentially enhanced the effect of ASD. Although the carbon sources used in that study are not available or cost-effective in the United States, locally available byproducts such as brewer's spent grain (BSG) could be used as an alternative, given that BSG mixed with yeast (*Saccharomyces cerevisiae*) can produce bioethanol [26].

BSG is the main byproduct of beer brewing. Although the composition of BSG may change depending on the operating conditions, BSG is generally rich in polysaccharides (cellulose and hemicellulose), proteins, and minerals [27]. The fermentable polysaccharides in BSG make it a potential resource for yeast fermentation [26]. Moreover, the anaerobic condition created during ASD is also aided by yeast, especially by facultatively anaerobic *S. cerevisiae* [28]. Liu et al. [29] evaluated the effects of ASD on weed control using several carbon sources mixed with ethanol and yeast, and found that yeast amendment enhanced the suppressing effect of ASD. However, just like applying other carbon sources in ASD with relatively large nitrogen concentrations, applying several tons of BSG to a field may release excess nitrogen into the environment. The excess nitrogen from ASD may also cause salt damage to the crop [15]. Moreover, excess nitrogen may cause cropping system issues such as excessive vegetative growth, increased lodging, delayed fruit maturity, increased insect and disease infestations, and enhanced weed growth [30]. Reducing the C input from ASD could not only mitigate these environmental impacts but also reduce material cost as well as the labor cost for applying C. However, research on determining optimal C rates for consistent ASD effects is ongoing, and no published results are available to suggest whether or not yeast addition could enhance ASD at a low C rate.

Annual hill plasticulture production is the most common system for strawberry production in Virginia and the mid-Atlantic region of the USA [31]. Strawberries are often planted consecutively on the same land with limited fallow time during the summer months to prepare the land and disinfest the soil before the next planting. Strawberry production is strongly hindered by weed competition, so pre-plant weed control is essential. A strawberry crop may yield less as a result of competition with annual weeds such as redroot pigweed (*Amaranthus retroflexus* L.) and common chickweed (*Stellaria media* L. Vill.); biennials including wild carrot (*Daucus carota* L.); and perennials including dandelion (*Taraxacum officinale* L. Weber ex F.H. Wigg.), white clover (*Trifolium repens* L.), and yellow nutsedge (*Cyperus esculentus* L.) [32]. A meta-analysis of 533 ASD experiments [9] showed that ASD suppressed yellow nutsedge but did not suppress redroot pigweed. Moreover, in most studies, weed suppression was observed at high soil temperatures (>35 °C) and with carbon source rates greater than 1 kg biomass m$^{-2}$.

Strawberries are highly susceptible to soilborne plant pathogens including *Pythium*, *Macrophomina*, *Fusarium*, and *Rhizoctonia* species. Infection by *Pythium* spp., including *Pythium irregulare*, is often a component of the black-root-rot-disease complex [33]. Black root rot is estimated to reduce strawberry yields by 20% to 40%, and for this reason, pre-plant soil fumigation is routinely practiced in the region [33]. The potential of using ASD as an alternative to traditional soil fumigation for strawberry has been reported [34,35]. Browne et al. [36] examined the effect of ASD on *Pythium ultimum* in Prunus replant disease, but no published research has investigated the effect of ASD on *P. irregulare*. Using ASD for *P. irregulare* control could increase the potential of ASD for strawberry soilborne disease control in general and would extend the potential spectrum of use for ASD. Recent research has evaluated ASD for strawberry and several other crops in other geographic sites in the USA [34,35]. Shennan et al. [37] indicated that ASD with RB controlled several pathogens, such as *V. dahilae*, *Fusarium oxysporum*, and *Pythium* spp., and also provided marketable strawberry yields which were equivalent to chemical fumigation. These studies showed that ASD could be a potentially viable alternative in strawberry for soil disinfestation. However, no ASD protocol has been developed for the conditions prevalent for strawberry growers in Virginia and the mid-Atlantic region. We hypothesized that yeast addition could enhance the effect of ASD on weeds and *P. irregulare*, and that yeast could improve the effect of ASD with reduced rates of C source. The objective of the first study was to evaluate the efficacy of BSG with and without yeast in ASD treatments for weed control and suppression of *P. irregulare*. The objective of the second study was to determine whether or not C dose rates could be reduced for ASD when using yeast.

## 2. Materials and Methods

### 2.1. Experiment Setup

Two greenhouse studies were initiated in a double-polyethylene-covered greenhouse at the Southern Piedmont Agricultural Research and Extension Center (AREC), Blackstone, VA, starting in April 2019. Greenhouse thermostats were set to begin cooling at 25 °C. In the greenhouse, 20 cm tall and 15 cm diameter custom-made containers [29] were used. For each pot treated with ASD, soil collected from the Southern Piedmont AREC (sandy loam, pH = 6.5, soil density =1.08 g/cm$^3$, 6.8 kg/container) was premixed in a tub with treatment-appropriate carbon sources. The soil in the containers was neither saturated nor flooded. Soil moisture content was adjusted to 20% using tap water. The 20% soil moisture content was the field capacity for the sandy loam soil [38]. Oxidation-reduction-potential (ORP) probes and temperature sensors were buried at a 15 cm depth. Black 1.25 mil virtually impermeable plastic film (VIF, Raven Industries Engineered Films Division, Sioux Falls, SD, USA) was secured on top of each pot except the non-treated control. The bottom of the containers was meticulously polished, resulting in an extremely narrow crevice between the containers' base and the sink when it was placed in a water basin. Consequently, the ingress of water was impeded. A black sheer voile fabric (Joann Fabric, Virginia Beach, VA, USA) was secured to the container bottom which acted as a permeable barrier and

allowed water movement from the tub upward into the container only through capillary action. Thus, when ASD treatments were initiated and the containers had been placed into trays with a water level of 10 cm, the soil in the containers would not be saturated or flooded, but the saturation level would be maintained during ASD through upward capillary action. The container top was sealed to minimize loss of water from evaporation from within the container, so there would be little need for replacement of water within the container (Figure 1). Those approaches were aimed at enhancing the capacity of the container to effectively mimic the field conditions, enabling more accurate simulation of open-field environments.

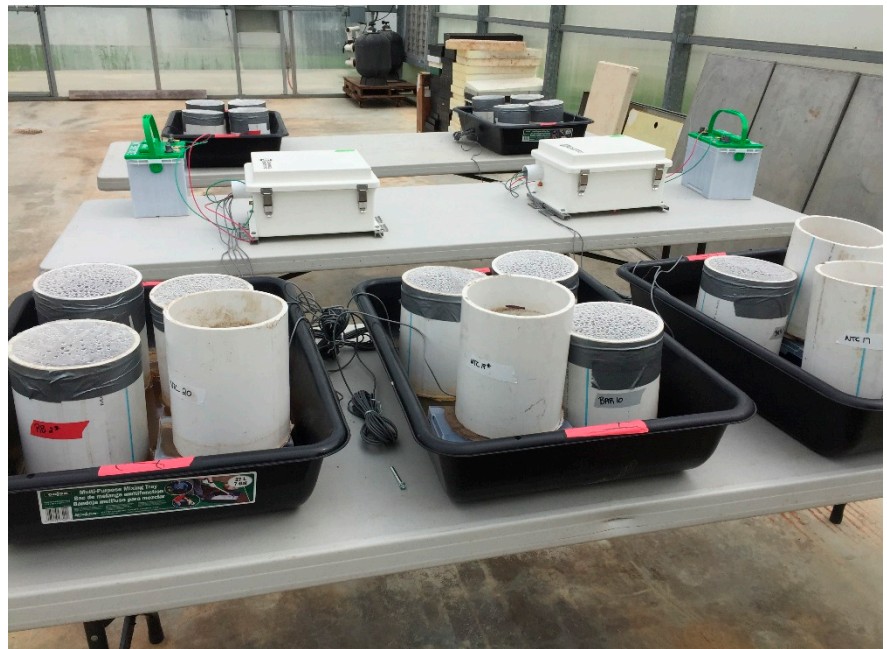

**Figure 1.** Overall experimental set up in the greenhouse at Blackstone, Virginia.

Carbon amendment rates and amounts were calculated based on the recommendations of Butler et al. [17]. The full dose carbon rate used in this study was 4 mg carbon/g soil or 16 t/ha. The full dose rate of carbon sources used was dry RB (63 g/container, 9.4 mg/g soil, 45% total C and 2.1% total N) and dry brewer's spent grain (BSG, 64 g/container, 9.4 mg/g soil, 44% total C and 3% total N, Commonwealth Brewing Company, Virginia Beach, VA, USA). To simulate ethanol fermentation in soil, distiller's yeast (Distiller's Active Dry Yeast, Red Star Yeast Co., Milwaukee, WI, USA) was mixed with carbon sources in appropriate treatments. The full dose rate of distiller's dry yeast was 0.06 g/container or 8.8 mg/g soil.

Each greenhouse study was performed with four replications in each of two repeated runs of the study. The containers were arranged in trays in a completely randomized design.

Study 1 compared the effects of BSG or RB, with or without yeast, to an untreated control, with or without yeast. The treatments were as follows in study 1: BSG, BSG + yeast, RB, RB + yeast, non-treated control and non-treated control + yeast. Study 1 was conducted from 17 April 2019 to 8 May 2019 and repeated from 6 June 2019 to 26 June 2019. Study 2 investigated the effects of BSG rate, with or without yeast, at reduced C rates, to an untreated control with or without yeast, and BSG at full rate involving the following treatments: BSG at full rate (64 g/container, 9.4 mg/g soil), BSG at half rate (32 g/container, 4.7 mg/g soil), BSG at half rate + yeast (0.03 g/container, 4.4 mg/g soil), BSG at one-third rate (21 g/container, 3.1 mg/g soil), BSG at one-third rate + yeast (0.02 g/container, 2.9 mg/g soil), non-treated control and non-treated control + yeast (0.06 g/container, 8.8 mg/g soil). Study 2 was first conducted from 18 July 2019 to 8 August 2019 and then repeated from 22 October 2019 to 13 November 2019.

## 2.2. Inoculum Preparation and Sensor Installation

In both studies, 100 common chickweed (*S. media*) seeds, 100 redroot pigweed (*A. retroflexus*) seeds, 100 white clover (*T. repens)* seeds, 10 yellow nutsedge tubers (*C. esculentus*), and inoculum consisting of 15 millet (*Urochloa ramosa* L.) seeds infected by *P. irregulare* were used in each container. All weeds were put in one inoculum bag (5.5 cm × 3 cm, nylon mesh, 20 μm pore size), and the millet seeds were put in another inoculum bag. The inoculum bags did not contain any soil, debris or substrate. Both bags were buried at approximately 12.5 cm above the container bottom. Common chickweed, redroot pigweed, and white clover were obtained from Herbiseed, Twyford, England. Yellow nutsedge tubers in study 1 were harvested from a local farm in Virginia Beach, VA, USA, and tubers in study 2 were harvested from the Hampton Roads Agricultural Research and Extension Center (AREC). The tubers used in both studies were approximately 1 cm in diameter and without abiotic or biotic damage.

The *P. irregulare* isolate (accession number OP 933289) was provided by co-author Dr. Xuemei Zhang at the Southern Piedmont Agricultural Research and Extension Center, Blackstone, VA. *P. irregulare* was cultured on potato dextrose agar (PDA). Millet seeds were prepared by soaking the seeds in deionized water for an hour (40 g millet seeds were soaked in 100 mL water), and then autoclaving the millet seeds at 121 °C for 45 min, once a day for three consecutive days. Inoculum was prepared by removing 5 mm diameter circular plugs from a PDA culture of the pathogen and mixing these with the sterilized millet seeds. The millet seeds and *P. irregular*-colonized agar pieces (8–10 pieces) were incubated at 25 °C in flasks in darkness for 2 weeks. Flasks were shaken by hand to ensure that PDA plugs were evenly distributed among the seeds. Approximately 15 colonized millet seeds were also plated onto a modified PARP medium (Sigma-Aldrich Chemie GmbH, Buchs, Switzerland) [36] at the same time that containers were inoculated to confirm that *P. irregulare* remained present in the inoculated seed. The modified PARP medium was made by combining cornmeal agar (17 g/L), Tween 20 (1 mL/L), Rose Bengal (25 μg/mL, a pink stain), rifampicin (10 μg/mL), ampicillin (250 μg/mL), pimaricin (5 μg/mL) and benomyl (40 μg/mL). The inoculum bag was prepared with nylon mesh fabric (2 mm sieve) and each bag contained 15 infested millet seeds.

Redox potential (Eh) sensors (ORP2000 Extended Life ORP Sensor, Sensorex, Garden Grove, CA, USA) were inserted at a 15 cm depth to evaluate soil anaerobic conditions. The ORP sensors were connected to a data-logging system (CR-1000, Campbell Scientific, Logan, UT, USA). Soil temperature sensors (U12 Deep Ocean Temperature Data Logger, Onset, Bourne, MA, USA) were buried at a 15 cm depth. Due to the limited number of sensors, half of the replicates per treatment were measured for redox potential and three of the four replicates per treatment were measured for soil temperature. Both sensors recorded readings every 10 min.

## 2.3. ASD Treatment Initiation and Post-Assessments

All of the containers were arranged in several water basins (76 cm × 40 cm) in a completely randomized design with four replicates. Each tray held six containers. The basins were filled with tap water and the water level was kept approximately 10 cm above the bottom of the containers. Water levels were maintained for the duration of each study, except for the non-treated controls. The non-treated containers were raised by wooden blocks and kept away from water in the basins. The non-treated controls were watered from the top once the soil dried. Each run of the study lasted 21 days and the viability of weed seeds was determined by a Tetrazolium chloride (TZ) assay [39]. The seeds of common chickweed, redroot pigweed and white clover were cut in half using a scalpel blade, and the TZ solution was dropped on both halves of each seed. The viability of yellow nutsedge tubers was determined by sprouting the tubers in a plastic nursery pot (10 cm diameter and 9 cm height), one treatment per container, five tubers per container, using BM 7 all-purpose bark mix (Berger, Laval, QC, Canada) in a growth chamber (23 °C, 16 h day length). The viability of *P. irregulare* was determined using a modified PARP

medium [36]. After ASD treatment, the recovered inoculum from *P. irregulare* inoculum bags was air-dried and ground using a sterile mortar and pestle. Then, ground samples (4 g) were added to 20 mL deionized water. The well-mixed sample solution (1 mL) was transferred to a modified PARP medium using an inoculating loop. The inoculated plates were incubated in the dark at 25 °C for 48 h. Macroscopically visible colonies of *P. irregulare* were counted 48 h after plating and reported as CFU $g^{-1}$ dry inoculum.

Soil samples (50 g/container) were collected at a 15 cm depth in study 1 on 8 May 2019 and 26 June 2019, and in study 2 on 8 August 2019 and 13 November 2019. The samples in study 1 were stored in 50 mL centrifuge tubes and then sent on ice bags to the University of Tennessee for determination of volatile fatty acids (VFAs) in soil. Due to funding limitations, the samples in study 2 could not be analyzed. Briefly, 30 g of water-saturated soil was extracted with 20 mL of 1 M KCl [40] for 30 min, centrifuged, and then filtered (0.2 μm). The external standard method was used to analyze the extracts for concentrations of acetic, propionic, *n*-butyric, isobutyric, valeric, and isovaleric acids by high-pressure liquid chromatography (HPLC) as described by Shrestha et al. [41].

### 2.4. Statistical Analyses

Data were checked first for normality and homogeneity of variance assumptions and were subject to transformation if they did not meet the assumptions. Data that met the normality assumptions were then analyzed using ANOVA and Fisher's protected least significant difference (LSD) test using JMP v. 14 (SAS Institute Inc., Cary, NC, USA). In study 1, carbon source, yeast addition and runs were treated as main effects for factorial analysis. The interactions among carbon source, yeast and runs were also evaluated. In study 2, due to the limitation of sensors, the treatment of the full BSG rate with yeast was absent; so, the carbon rate and yeast were not treated as separate main effects. Thus, the treatments and repeated trial were treated as main effects. The treatment by run interaction was evaluated. When the interaction effect was not significant, only main effects were analyzed. When the interaction effect was significant, a contrast test was used to compare each carbon source with yeast or without yeast, using Tukey's Honestly Significant Difference. The temperature data were recorded for the 21-day duration of the ASD treatments; maximum and minimum temperatures achieved during the treatment period were also recorded. The cumulative soil anaerobic condition [17] was calculated based on the hourly average redox potential (Eh). The absolute value of the difference between each hourly average Eh and the calculated critical redox potential (CEh) was summed up over the whole three-week ASD period. The critical redox potential was calculated using the formula Ceh = 595 mV − 60 mV × soil pH [42]. Five volatile fatty acids were identified, and two-way ANOVA was conducted for each acid. Carbon source and yeast addition were treated as fixed effects and runs as a random effect.

## 3. Results

### 3.1. Weed and P. irregulare Viability

The carbon by yeast by run, carbon by run and yeast by run interactions were not significant in study 1 (Table 1). The interaction effects between carbon source and yeast amendment were significant for redroot pigweed, white clover and *P. irregulare* (Table 1). Effects of carbon source in study 1 were significant for all weed seeds, *P. irregulare* and cumulative anaerobicity, while the effects of adding yeast were significant for all weeds except yellow nutsedge and for *P. irregulare*, but not for cumulative anaerobicity (Table 1). Both BSG and RB, with or without yeast, reduced the seed viability of redroot pigweed, white clover and *P. irregulare* compared to the non-treated control (Table 2). BSG, with or without yeast, reduced seed viability of redroot pigweed more than RB, with or without yeast, while BSG with yeast reduced the viability of white clover seed more than RB with yeast; white clover seed viability with BSG and yeast was similar to that for RB without yeast. BSG with yeast reduced the numbers of CFU of *P. irregulare* more than RB, with or without yeast (Table 2). Yeast addition did not increase the effectiveness of RB but

increased BSG effectiveness against pigweed, white clover and *P. irregulare.* Regardless of the carbon source, ASD treatments reduced common chickweed and yellow nutsedge viability compared to the non-treated control (Table 3).

**Table 1.** Significance (*p*-value) from a two-way ANOVA evaluation of viability of weeds and the colony count of *P. irregulare* after treatment with different carbon sources and yeast in study 1.

|  | Common Chickweed | Redroot Pigweed | White Clover | Yellow Nutsedge | *P. irregulare* | Anaerobic Condition |
|---|---|---|---|---|---|---|
|  | Probability > F | | | | | |
| Carbon source | <0.0001 | <0.0001 | <0.0001 | 0.0133 | <0.0001 | 0.0018 |
| Yeast | 0.0007 | 0.0005 | 0.0057 | 0.8347 | 0.0317 | 0.3494 |
| Run | 0.092 | 0.11 | 0.289 | 0.730 | 0.264 | 0.182 |
| Carbon × yeast | 0.1290 | 0.0054 | 0.0033 | 0.8815 | 0.0122 | 0.8142 |
| Carbon × run | 0.837 | 0.940 | 0.089 | 0.077 | 0.518 | 0.235 |
| Yeast × run | 0.342 | 0.710 | 0.789 | 0.485 | 0.127 | 0.508 |
| Carbon × yeast × run | 0.777 | 0.852 | 0.823 | 0.613 | 0.921 | 0.617 |

**Table 2.** Viability of redroot pigweed and white clover seed and colony forming units of *P. irregulare* after treatment with two different carbon sources, with or without yeast, in study 1.

| Carbon Source [a] | Yeast | Redroot Pigweed | | White Clover | |
|---|---|---|---|---|---|
|  |  | Seed Viability (%) | Contrast *p*-Value [c] | Seed Viability (%) | Contrast *p*-Value [c] |
| Brewer's spent grain | Without | 27.0 b [b] | <0.0001 | 21.0 b | <0.0001 |
|  | With | 15.0 d |  | 11.0 d |  |
| Rice bran | Without | 23.0 c | 0.30 | 13.0 cd | 0.30 |
|  | With | 20.0 c |  | 15.0 c |  |
| Non-treated control | Without | 74.0 a | 0.15 | 82.0 a | 0.07 |
|  | With | 68.0 a |  | 78.0 a |  |
|  |  | *P. irregulare* | | | |
| Carbon Source | Yeast | Colony forming units | Contrast *p*-value [c] | | |
| Brewer's spent grain | Without | 51.3 b | <0.0001 | | |
|  | With | 28.0 c |  | | |
| Rice bran | Without | 53.9 b | 0.84 | | |
|  | With | 52.3 b |  | | |
| Non-treated control | Without | 164.4 a | 0.42 | | |
|  | With | 172.4 a |  | | |

[a] The rates of carbon sources and yeast/container were as follows: brewer spent grain 64 g/container, rice bran 63 g/container, yeast 0.06 g/container. [b] Means followed by different letters within a column of each inoculum (*n* = 8) are statistically different using Fisher's protected least significant difference (LSD) test at $p \leq 0.05$. [c] Contrast yeast *p*-value was the *p*-value for contrast tests comparing each carbon source with yeast to without yeast, using Tukey's Honestly Significant Difference (HSD) at $\alpha = 0.05$.

**Table 3.** Effect of carbon sources on viability of common chickweed, yellow nutsedge, and cumulative anaerobicity after treatment with different carbon sources in two runs of study 1.

| Carbon Sources [a] | Common Chickweed Viability (%) | Yellow Nutsedge Viability (%) | Anaerobic Condition (Vh) |
|---|---|---|---|
| Brewer's spent grain | 17.5 b [b] | 0.0 b | 179.8 a |
| Rice bran | 20.9 b | 1.3 b | 120.7 b |
| Non-treated control | 69.3 a | 72.5 a | 4.7 c |

[a] The rates of carbon sources and yeast/container were as follows: brewer spent grain 64 g/container, rice bran 63 g/container, yeast 0.06 g/container. [b] Means (*n* = 16) followed by different letters within a column are statistically different using Fisher's protected least significant difference (LSD) test at $p \leq 0.05$.

The treatment by run interaction was not significant in study 2, in which use of BSG with or without yeast reduced the seed viability of all weeds and CFUs of *P. irregulare* in both experiments (Table 4). All rates of BSG lowered the viability of all weeds significantly, as well as the CFU of *P. irregulare*, compared to the non-treated control with and without yeast. Seed viability of common chickweed and CFU of *P. irregulare* were lowest when the full rate of BSG was used compared to the use of only one-third or one-half of the full rate of BSG. Seed viability of redroot pigweed, white clover and yellow nutsedge was lowest when the full rate of BSG had been applied (without yeast), and the one-third BSG and one-half BSG with yeast treatments. The addition of yeast to the one-half and one-third BSG treatments significantly reduced the viability of common chickweed, redroot pigweed, and white clover at a given reduced carbon dose rate. The viability of *P. irregulare* was reduced with yeast at half the BSG rate compared to the same C rate without yeast (Table 4).

**Table 4.** Results from a two-way ANOVA evaluation of viability of weeds, the colony count of *P. irregulare* and cumulative anaerobicity treatments with difference carbon rates, and with or without yeast in study 2.

| Treatment [a] | Weed Viability (%) | | | | *P. irregulare* (CFU) [b] | Cumulative Anaerobicity (Vh) |
|---|---|---|---|---|---|---|
| | Common Chickweed | Redroot Pigweed | White Clover | Yellow Nutsedge | | |
| Brewer's spent grain full | | | | | | |
| Without yeast | 16.6 d [c] | 18.9 c | 24.8 c | 2.5 c | 46.3 e | 320.4 a |
| Brewer's spent grain half | | | | | | |
| Without yeast | 31.2 b | 47.5 b | 46.8 b | 8.8 c | 87.5 b | 250.9 ab |
| With yeast | 21.3 c | 19.2 c | 25.0 c | 8.8 c | 70.0 c | 223.5 ab |
| Brewer's spent grain one-third | | | | | | |
| Without yeast | 32.6 b | 44.4 b | 43.9 b | 20.0 b | 68.8 cd | 138.0 b |
| With yeast | 23.5 c | 21.8 c | 24.6 c | 6.3 bc | 61.2 d | 254.9 ab |
| Non-treated control | | | | | | |
| Without yeast | 69.9 a | 72.9 a | 76.7 a | 72.5 a | 186.9 a | 20.1 c |
| With yeast | 74.6 a | 74.8 a | 74.5 a | 66.3 a | 147.5 a | 76.0 c |
| *p* value | | | | | | |
| Treatment | <0.0001 | <0.0001 | <0.0001 | <0.0001 | <0.0001 | <0.0001 |
| Run | 0.637 | 0.637 | 0.060 | 0.219 | 0.105 | 0.324 |
| Treatment × run | 0.402 | 0.242 | 0.140 | 0.271 | 0.922 | 0.450 |

[a] The rates per container of C sources and yeast were as follows: brewer's spent grain full rate 64 g, brewer's spent grain half rate 32 g, half-rate yeast 0.03 g, brewer's spent grain one-third rate 21 g, one-third rate yeast 0.02 g, non-treated control with yeast 0.06 g. There was no full rate BSG with yeast treatment for this variable. [b] CFU = colony forming units. [c] Means (*n* = 8) followed by different letters within a column of each inoculum are statistically different using Fisher's protected least significant difference (LSD) test at $p \leq 0.05$.

### 3.2. Cumulative Soil Anaerobicity

Adding either BSG or RB increased cumulative anaerobicity compared to the non-treated control, and BSG addition resulted in higher anaerobicity than RB (Table 3). All rates of BSG increased cumulative anaerobicity significantly more than the non-treated control, with or without yeast addition. Higher rates of BSG resulted in numerically higher anaerobicity (Table 4), but the differences among different carbon rates were not significant, except for the one-third BSG rate without yeast. The full rate of BSG with or without yeast produced significantly higher cumulative anaerobicity than the one-third rate of BSG without yeast. Yeast amendment did not significantly increase anaerobicity

from RB in study 1 (the only study in which it was tested) or from BSG in either studies (Tables 1 and 4).

### 3.3. Temperature

The average, minimum and maximum soil temperatures within the containers did not differ among treatments during the ASD period (Tables 5 and 6). Soil temperatures at a 15 cm depth generally ranged from 17 °C to 43 °C in study 1, and 20 °C to 43 °C in study 2. There were no significant differences in average temperature among treatments in either study. The mean temperature for all treatments in study 1 was approximately 26 °C, and 29 °C in study 2.

**Table 5.** Mean, minimum and maximum temperatures over three-week period of anaerobic soil disinfestation (ASD) process with several different C sources and yeast amendment in study 1 at 15 cm depth averaged over two repeated trials.

| Treatment [a] | Temperature (°C) ± Standard Deviation | | | | | |
|---|---|---|---|---|---|---|
| | Mean for Three Weeks | Mean for Week 1 | Mean for Week 2 | Mean for Week 3 | Minimum for Three Weeks | Maximum for Three Weeks |
| Brewer's spent grain full | | | | | | |
| Without yeast | 25.9 ± 0.5 | 25.3 ± 0.4 | 25.5 ± 0.5 | 27.1 ± 0.6 | 17.4 ± 2.0 | 42.2 ± 0.7 |
| With yeast | 26.6 ± 0.3 | 25.8 ± 0.3 | 26.0 ± 0.5 | 28.0 ± 0.3 | 18.1 ± 0.7 | 43.3 ± 0.5 |
| Rice bran | | | | | | |
| Without yeast | 26.3 ± 0.3 | 25.7 ± 0.3 | 25.7 ± 0.5 | 27.7 ± 0.3 | 17.8 ± 0.7 | 43.1 ± 0.5 |
| With yeast | 26.6 ± 0.3 | 25.9 ± 0.2 | 26.1 ± 0.4 | 28.0 ± 0.2 | 17.5 ± 0.4 | 41.2 ± 0.4 |
| Non-treated control | | | | | | |
| Without yeast | 25.7 ± 0.4 | 24.9 ± 0.4 | 25.8 ± 0.6 | 26.4 ± 0.5 | 18.2 ± 0.7 | 41.7 ± 0.7 |
| With yeast | 26.1 ± 0.4 | 25.2 ± 0.3 | 25.7 ± 0.5 | 27.4 ± 0.3 | 17.3 ± 0.6 | 40.2 ± 0.5 |
| *p*-value [b] | | | | | | |
| Carbon source | 0.282 | 0.100 | 0.944 | 0.059 | 0.277 | 0.606 |
| Yeast | 0.161 | 0.217 | 0.501 | 0.074 | 0.300 | 0.964 |
| Carbon × yeast | 0.892 | 0.834 | 0.858 | 0.660 | 0.325 | 0.240 |

[a] The rates of C sources and yeast were as follows: brewer spent grain 64 g, rice bran 63 g, yeast 0.06 g. [b] The run effect and the interactions with run effect were not significant, and the *p*-value of the run effect and interactions with run effect were not presented.

### 3.4. Volatile Fatty Acids

Five volatile fatty acids (VFAs) were identified as follows: acetic acid (AA), propionic acid (PA), isobutyric acid (IBA), n-butyric acid (nBA), and isovaleric acid (IVA). A two-way ANOVA was conducted for each acid, and carbon and yeast were treated as main effects (Table 7). Concentrations of all five VFAs varied significantly among carbon sources (Table 7). Use of RB as a carbon source for ASD did not increase VFA concentrations over the non-treated control, regardless of yeast, but concentrations of all five VFAs were higher when BSG was used as a carbon source compared to use of RB or the non-treated control, whether or not yeast had been added (Table 8). Use of BSG as a carbon source was associated with higher PA concentrations whether amended with yeast or not (Table 8). A significant interaction between carbon source and yeast amendment treatments was observed for PA, but not for the other four VFAs. Thus, a contrast was used to compare PA concentrations among carbon-source treatments with or without yeast. Yeast amendment increased PA concentrations significantly when BSG was used as a source of carbon, but not for RB treatments or the non-treated control (Table 8).

**Table 6.** Mean, minimum and maximum temperatures over three-week period of anaerobic soil disinfestation (ASD) process with several different C sources and yeast amendment in study 2 at 15 cm depth averaged over two runs.

| Treatment [a] | Temperature (°C) ± Standard Deviation | | | | | |
|---|---|---|---|---|---|---|
| | Mean for Three Weeks | Mean for Week 1 | Mean for Week 2 | Mean for Week 3 | Minimum for Three Weeks | Maximum for Three Weeks |
| Brewer's spent grain full | | | | | | |
| Without yeast | 28.7 ± 0.2 | 29.2 ± 0.3 | 29.5 ± 0.2 | 27.1 ± 0.1 | 19.5 ± 0.7 | 41.5 ± 1.2 |
| Brewer's spent grain half | | | | | | |
| Without yeast | 29.5 ± 0.6 | 30.0 ± 0.7 | 30.5 ± 0.4 | 27.9 ± 0.8 | 19.3 ± 2.1 | 43.9 ± 1.1 |
| With yeast | 30.0 ± 0.6 | 30.6 ± 0.6 | 30.6 ± 0.4 | 28.7 ± 0.8 | 18.9 ± 0.9 | 44.5 ± 1.2 |
| Brewer's spent grain one-third | | | | | | |
| Without yeast | 29.4 ± 0.6 | 30.0 ± 0.8 | 30.2 ± 0.3 | 28.0 ± 0.7 | 18.8 ± 0.4 | 43.1 ± 1.2 |
| With yeast | 29.4 ± 0.6 | 30.1 ± 0.6 | 30.1 ± 0.6 | 28.0 ± 0.8 | 19.1 ± 0.7 | 42.5 ± 1.0 |
| Non-treated control | | | | | | |
| Without yeast | 28.7 ± 0.7 | 29.2 ± 0.8 | 29.5 ± 0.7 | 27.1 ± 0.8 | 19.5 ± 1.6 | 41.5 ± 1.3 |
| With yeast | 28.5 ± 0.5 | 28.9 ± 0.8 | 29.2 ± 0.3 | 27.0 ± 0.7 | 20.9 ± 1.1 | 40.4 ± 0.1 |
| *p*-value [b] | | | | | | |
| Treatment | 0.384 | 0.344 | 0.294 | 0.375 | 0.176 | 0.854 |

[a] The rates of C sources and yeast were as follows: brewer's spent grain full rate 64 g, brewer's spent grain half rate 32 g, half rate yeast 0.03 g, brewer's spent grain one-third rate 21 g, one-third rate yeast 0.02 g, non-treated control with yeast 0.06 g. There was no full rate BSG with yeast treatment for this variable. [b] The treatment by run effect and the run effect were not significant, and the *p*-value of the run effect and interactions with run effect were not presented.

**Table 7.** Significance of carbon source and yeast amendment effects on five volatile fatty acids in two-way ANOVAs in study 1.

| | Acetic Acid | Propionic Acid | Isobutyric Acid | *n*-Butyric Acid | Isovaleric Acid |
|---|---|---|---|---|---|
| | Probability > F | | | | |
| Carbon | <0.0001 | <0.0001 | <0.0001 | <0.0001 | 0.0004 |
| Yeast | 0.056 | 0.0021 | 0.646 | 0.090 | 0.091 |
| Run | 0.100 | 0.142 | 0.452 | 0.066 | 0.065 |
| Yeast × Carbon | 0.715 | 0.0053 | 0.378 | 0.489 | 0.795 |
| Carbon × run | 0.108 | 0.928 | 0.076 | 0.622 | 0.462 |
| Yeast × run | 0.813 | 0.062 | 0.066 | 0.360 | 0.775 |
| Carbon × yeast × run | 0.250 | 0.094 | 0.662 | 0.965 | 0.278 |

**Table 8.** Volatile fatty acids concentrations after treatment completion with different carbon sources and yeast in study 1.

| Inoculum | Carbon Sources [a] | Concentration (mmol kg$^{-1}$ of Soil) | | |
|---|---|---|---|---|
| | | With Yeast | Without Yeast | Contrast *p*-Value |
| Acetic acid | Brewer's spent grain | 1.323 a [b] | 1.436 a | - |
| | Rice bran | 0.430 b | 0.166 b | - |
| | Non-treated control | 0.203 b | 0.037 b | - |
| Propionic acid | Brewer's spent grain | 0.152 a | 0.077 a | <0.0001 [c] |
| | Rice bran | 0.024 b | 0.024 b | 0.84 |
| | Non-treated control | 0.017 b | 0.009 b | 0.43 |

**Table 8.** *Cont.*

| Inoculum | Carbon Sources [a] | Concentration (mmol kg$^{-1}$ of Soil) | | |
|---|---|---|---|---|
| | | **With Yeast** | **Without Yeast** | **Contrast *p*-Value** |
| Isobutyric acid | Brewer's spent grain | 0.117 a | 0.161 a | - |
| | Rice bran | 0.033 b | 0.014 b | - |
| | Non-treated control | 0.000 b | 0.001 b | - |
| *n*-butyric acid | Brewer's spent grain | 0.390 a | 0.483 a | |
| | Rice bran | 0.118 b | 0.005 b | - |
| | Non-treated control | 0.070 b | 0.004 b | - |
| Isovaleric acid | Brewer's spent grain | 0.472 a | 0.427 a | - |
| | Rice bran | 0.166 b | 0.063 b | - |
| | Non-treated control | 0.043 b | 0.002 b | - |

[a] The rates of carbon sources and yeast/container were as follows: brewer spent grain 64 g, rice bran 63 g, yeast 0.06 g. [b] Means followed by different letters within a column of each acid are statistically different using Fisher's protected least significant difference (LSD) test at $p \leq 0.05$. N = 16 for all acids except for propionic acid $n = 8$. [c] Contrast *p*-value was the *p*-value using Tukey's Honestly Significant Difference (HSD) at $\alpha = 0.05$., for comparing yeast vs. no yeast for each carbon source. The contrast was only conducted when carbon sources by yeast interaction were significant.

## 4. Discussion

ASD treatments using BSG provided good control of common chickweed, redroot pigweed, white clover and yellow nutsedge in this study, although additional research under field conditions is warranted. Khakda et al. [43] reported that ASD field treatments reduced the biomass of barnyard grass (*Echinochloa crus-galli* (L.)) and several other grasses and broadleaf weed species with various local carbon sources such as molasses, goat manure and lentil husks. Yellow nutsedge is a problematic perennial weed species, in plasticulture systems. Another ASD study using wheat bran as a carbon source demonstrated lower yellow nutsedge sprouting and reproduction rates under both controlled and field conditions [35]. Redroot pigweed also showed moderate sensitivity to ASD treatment with 50% mortality achieved under field conditions [43].

In this study, BSG at three different dose rates reduced seed viability of common chickweed, redroot pigweed, white clover, and tuber viability in yellow nutsedge, and the survival of *P. irregulare* compared to the non-treated control. Moreover, BSG with and without yeast inoculation had effects comparable to RB. Weed suppression similar to our study was also reported by other researchers for common chickweed seed [44] and yellow nutsedge tubers [2,9]. However, the reduction of redroot pigweed seen in our work was not consistent with results from another study [14]. The main difference between our studies and the cited literature was the carbon sources used, which may affect weed-control efficacy. For example, our studies used BSG as a carbon source, while the cited studies [2,35,44,45] used carbon sources such as cover crops (i.e., *Brassica juncea*, *Sinasis alba*, *Eruca sativa*, and *Secale cereale*), molasses, or a green manure crop (*Festuca perennis* Lam.). The different moisture, texture, mixture, and rates of carbon sources may have resulted in differences in weed control. There is no other research on the effect of ASD on *P. irregulare*, while there are several research reports on other *Pythium* species and *Phytophthora*, such as *P. intermedium* [46], *P. ultimum* [36,47] and *Phytophthora nicotianae* [19]. The results of van Os et al. [46] indicated that microbial volatiles played an important role in suppressing pathogens such as *Pythium*, which may explain *Pythium* suppression in our studies. Additional research is needed on the mechanisms of how ASD suppresses pathogens via microorganism activities or by their metabolites, such as VFAs.

Although the precise mechanism of ASD is not fully understood, VFAs have been detected in various ASD studies, and have been reported as a critical component of control for some soilborne pathogens [7,16,48]. The VFAs could become lethal to many soilborne pathogens, which could be due to their ability to readily move across cell membranes, acidifying the cell cytoplasm [49]. In our study, multiple VFAs were generated during

ASD, similar to previous literature [7,35]. However, amending BSG with yeast did not increase VFA concentrations. One possible explanation could be that other organisms compete for labile carbon, and those organisms produce fewer VFAs due to metabolizing less carbon. As ASD is initiated, organisms such as *Clostridium*, *Enterobacter*, and *Acetobacter* rapidly reproduce and break down carbon sources into VFAs, alcohols, and $CO_2$ [16]. The competition between yeast and other microorganisms may occur at this stage. After that, products such as VFAs could be utilized by facultative anaerobic organisms such as *Bacillus* species, which have been reported as biocontrol organisms [7,50]. Yeast may enhance alcohol production and provide more labile carbon for such biocontrol organism growth. However, while the yeast amendment enhanced weed control in our research, it did not enhance VFA concentrations, indicating that the VFA concentrations may not be the predominant mechanism for weed suppression. Biological effects may directly affect weed seeds and tubers, but more research is needed to verify such conclusions. BSG had the potential to produce relatively high concentrations of volatile fatty acids, and yeast amendment could enhance the acids produced when mixed with BSG. However, BSG and yeast could be affected by environmental factors and result in inconsistent acid production. Thus, more research needs to be conducted to investigate how environmental factors influence the effect of BSG and yeast on acids.

Shrestha et al. [35] showed that yellow nutsedge viability changes may differ based on the carbon–nitrogen ratio or soil depth. Compared to the higher carbon–nitrogen-ratio treatments (40:1), application of lower carbon–nitrogen-ratio treatments (10:1) in ASD resulted in lower yellow nutsedge sprouting and reproduction rates. Previous research [15] also showed that the emergence of yellow nutsedge tubers was greater at a 5 cm depth compared to a 15 cm depth with ASD. Muramoto et al. [16] showed yellow nutsedge tubers at a 2 cm depth (where the soil is slightly drier) had higher emergence rates with ASD compared to tubers at a 15 cm depth. It is possible that the soil moisture and anaerobicity are different at different soil depths, which may lead to various ASD product concentrations at different soil depths. Such variable concentrations and anaerobicity may change the weed or pathogen-suppression effect. Combining those research studies and our studies, we could hypothesize that the lack of oxygen, along with increased anaerobic by-products, and anaerobic microbial activity at greater depths, may increase tuber mortality, even though yellow nutsedge tubers have a high flooding tolerance [51]. More research on the distribution of ASD products, and related soil physical and chemical parameters at different soil depths is needed to better understand ASD behavior in soil.

In our studies, ASD treatments resulted in a moderate mean soil temperature (25 °C and 30 °C) compared to a higher temperature (>35 °C) observed in other studies [17]. However, a meta-analysis by Shrestha et al. [9] showed that the effect of ASD for weed control required higher temperatures (>35 °C). There are several hypotheses for effective weed control at low or moderate temperatures. In many reported studies, the containers were flooded only at the beginning of the ASD period to create the anaerobic condition; in our study, the containers were immersed in water for 21 days, in order to maintain anaerobic conditions in the bioreactor. Water would have entered containers through the fabric-covered bottom opening, which means that soil temperatures might have been moderated in our studies by the immediate environment surrounding the container. The soil moisture level may have mitigated the soil temperature changes. Another explanation could be that the much greater anaerobic conditions promoted weed suppression under moderate temperature. Compared to some studies that observed effective weed control at higher temperatures (>35 °C) [52,53], this study found relatively higher anaerobic conditions (100–300 Vh versus 40–50 VH). While a low oxygen condition was generated in our studies, similar to other research using BSG [47], there was no strong correlation between weed control and anaerobic conditions, as with other studies [17,45]. This lack of correlation indicates that strong anaerobicity may not be a direct factor of weed control. Interactions among soil moisture, soil temperature, and weed suppression in ASD need to be studied under field conditions.

The use of ASD may be a more beneficial alternative to fumigants or synthetics herbicides in small farms or lower-resourced farming systems than in large-scale or mechanized farming systems. Poor weed management can result in crop losses and threaten food security in less developed regions [54]. Most growers in less developed areas may use intensive tillage or hand-weeding as a tool to manage weeds, but the high labor costs may reduce profits [55,56]. Additionally, improper tillage could also reduce soil moisture and soil quality traits such as organic-matter level [57]. Implementing ASD on small farms could also increase soil fertility as well as reduce the labor associated with weeding. Smallholder farms could also select locally available carbon sources such as agriculture byproducts and waste, manure and cover crops to offset the cost of ASD.

The use of ASD suppressed all four weed species and *Pythium* in the present study, and distiller's dry yeast was observed to enhance weed suppression resulting from ASD. Compared to the material cost of soil fumigants, US $1000–1800/ha [58], the material cost for BSG applied with yeast is much lower—BSG is currently free, while the cost of yeast may be $145/ha. However, the material cost for BSG would likely increase if demand increases significantly. The costs associated with the transportation and application of large volumes of carbon sources could also make ASD less economical, but using yeast to reduce the carbon dose rates necessary could mitigate such considerations. Brewer's spent grain contains organic nitrogen which might also substitute for the application of synthetic fertilizer. For example, the full rate BSG at 5 Mg/ha could provide approximately 350 kg/ha nitrogen. The general recommended preplant nitrogen application rate for strawberry is 68 kg/ha, which is far less than the nitrogen from BSG. The half rate of BSG could also provide nearly 125 kg/ha nitrogen. Although the nitrogen from BSG would not be fully decomposed, the preplant nitrogen fertilizer might not be necessary if only 20% of the nitrogen supplied by a full rate of BSG were available during the first five months, offsetting the cost of ASD by approximately $100/ha. Additionally, BSG also contains other nutrients, such as phosphorus, and calcium [27]. The use of BSG could potentially reduce fertilizer cost or increase soil fertility [59]. Thus, further research on the behavior of BSG at reduced rates in field conditions is necessary.

Rice bran has higher cellulose content than BSG (Table 9), and yeast primarily converts simple sugars, primarily glucose, to carbon dioxide and ethanol. Brewer's spent grain contains large amounts of fermentable sugars, making it an attractive carbon source for yeast [60]. Rice bran is low in sugar relative to grain-based carbon sources. Additionally, the difference in cellulose content between the two carbon sources may be attributed to the preference of yeast for BSG over RB. An acid or enzymatic pre-treatment would be needed to enhance the conversion of cellulose to glucose [27].

**Table 9.** The total carbon, total nitrogen and represented compounds comparison for Brewer's spent grain and rice bran.

|  | Brewer's Spent Grain | Rice Bran |
|---|---|---|
| Total C (% dry wt) | 44.8 * | 41.7 |
| Total N (% dry wt) | 3.8 | 3.1 |
| C/N ratio | 12.0 | 13.3 |
| Cellulose (% dry wt) | 17–25 | 30–35 |
| Hemicellulose (% dry wt) | 20–30 | 20–22 |
| Lignin (% dry wt) | 12–27 | 7–10 |
| Protein (% dry wt) | 15–24 | 11–17 |

* The rates in the table were from studies [27,59–64].

Given the beneficial results observed in this project from the application of yeast to reduce the dosage of carbon sources needed for effective ASD, future research is needed to refine techniques of yeast application, such as by adjusting the rate of yeast, priming yeast before application, or pre-treating carbon sources to increase their nutritional suitability for yeast. Future research is needed to clarify the short-term and long-term impacts of ASD

on soil nutrient availability, as well as whether or not nutrients from ASD carbon sources impact strawberry growth, canopy development and the timing and dynamics of flowering, or fruit set. Meanwhile, the economic analysis of ASD for large-scale field production is also essential.

**Author Contributions:** Conceptualization, D.L., J.S., C.J. and D.M.B.; methodology, J.S., C.J. and D.L.; software, D.L. and J.S.; validation, D.L., J.S., C.J., D.M.B. and J.D.; formal analysis, D.L., J.S., and C.J.; investigation, D.L., J.S., C.J., D.M.B. and X.Z.; resources, J.S., C.J., D.M.B., X.Z. and J.D.; data curation, D.L., J.S. and D.M.B.; writing—original draft preparation, D.L.; writing—D.L., J.S., C.J., D.M.B., J.D. and X.Z.; visualization, J.S. and D.L.; supervision, J.S.; project administration, J.S.; funding acquisition, J.S., C.J. and D.M.B. All authors have read and agreed to the published version of the manuscript.

**Funding:** This research was funded by Southern Region Small Fruit Consortium, North American Strawberry Growers Association and USDA NIFA Hatch Project VA-160077.

**Data Availability Statement:** All available data is presented in the publication.

**Acknowledgments:** The authors would like to thank Ned Jones, Robert Irby, Chuanxue Hong, Mark Hoffmann and Patricia Richardson for their technical support.

**Conflicts of Interest:** All authors declare no conflict of interest related to the publication of this manuscript.

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
