# Peer review of "Brewer’s Spent Grain with Yeast Amendment Shows Potential for Anaerobic Soil Disinfestation of Weeds and Pythium irregulare"

_agronomy, doi:10.3390/agronomy13082081_

Round 1
Reviewer 1 Report
The research (Brewer`s Spent Grain with Yeast Amendment Shows Potential for Anaerobic Soil Disinfestation of Weeds and Pythium irregulare) investigated the impact of brewer's spent grain (BSG), a locally available carbon source, on various weed species and the oomycete pathogen Pythium irregulare in ASD. Although the research idea is good there is some issue that needs to handle to validate the experimental and the results.
1. The authors need to add the greenhouse environmental conditions and also the chemical analysis of the used soil.
2. Did the authors sterilize the soil? If not, they must include microbial analysis of the used soil otherwise the design will have confused data.
3. The authors need to include the chickweed, redroot pigweed, and white clover seed sources and varieties numbers.
4. Did the authors sterilize the seeds before using them to valid the results?
5. The genetic identification of Pythium irregulare needs to be included with the accession number.
6. Inserting plant photos is recommended in the manuscript.
7. At the experiment ended did the authors examine the microbial content of the soil, more severe and pathogenic isolates could arise and damage the next growing plants.
8. The anaerobic soil microorganisms must include in the study to explain the experimental results.
English editing and grammar correction is recommended
Reviewer 2 Report
The authors showed that anaerobic soil disinfestation (ASD) using brewer’s spent grain (BSG) as a material was effective in controlling various weeds and phytopathogenic Pythium irregulare. In addition, it was clarified that when distiller’s yeast is added at the same time as BSG, the same disinfestation effect can be seen even if the amount of BSG added is reduced. These results are considered to be effective for the spread of ASD technique using locally produced BSG in the future.
The addition of yeast is effective with ASD using BSG, but not with rice bran (RB) (Table 2). The reason for this is thought to be the difference in composition. Could you show the difference between BSG and RB (e.g. by using a new table) as much as possible? For example (in addition to total carbon content and total nitrogen content), C/N ratio, pH, content of sugars, starch, cellulose, hemicellulose, protein, mineral (phosphate, potassium), etc.
Similarly, if authors are conducting a survey of the composition (total carbon content, total nitrogen content, etc.) of the soil used, could you please describe it?
Distiller’s yeast is thought to be able to utilize a limited range of components of BSG (such as sugars and amino acids) and convert them into other active components (such as ethanol, organic acids, and esters). Could you please describe why the same (or similar) disinfestation effect can be obtained by adding yeast even if the amount of BSG added is reduced, based on the difference in composition between BSG and RB?
The authors reported that the addition of yeast does not increase the amount of VFAs, which are thought to be active components of ASD (Table 8). If so, what kind of metabolites (by yeast) might be involved in weed germination inhibition?
In the ASD test with BSG, the remaining amount of Pythium irregulare was significantly reduced (Table 4). However, it is thought to be insufficient to prevent disease caused by this pathogen (I think that the pathogen should be reduced to the limit of detection). What measures do the authors think need to be taken hereafter to control this pathogen?
Please check the references (no. 3, 9, 22, 24, 29-31, 33, 43, 45, 46, 51, 55, 56, 59) (line 512-662).
